# Nature-Inspired Bioactive Compounds: A Promising Approach for Ferroptosis-Linked Human Diseases?

**DOI:** 10.3390/molecules28062636

**Published:** 2023-03-14

**Authors:** Sarah El Hajj, Laetitia Canabady-Rochelle, Caroline Gaucher

**Affiliations:** 1Université de Lorraine, CITHEFOR, F-54505 Vandoeuvre Les Nancy, France; 2Université de Lorraine, CNRS, LRGP, F-54000 Nancy, France; 3Université de Lorraine, CNRS, IMoPA, F-54000 Nancy, France

**Keywords:** ferroptosis, lipid peroxidation, iron chelation, bioactive compounds, cardiovascular diseases, neurodegenerative diseases

## Abstract

Ferroptosis is a type of cell death driven by iron overload and lipid peroxidation. It is considered a key mechanism in the development of various diseases such as atherosclerosis, Alzheimer, diabetes, cancer, and renal failure. The redox status of cells, such as the balance between intracellular oxidants (lipid peroxides, reactive oxygen species, free iron ions) and antioxidants (glutathione, glutathione Peroxidase 4), plays a major role in ferroptosis regulation and constitutes its principal biomarkers. Therefore, the induction and inhibition of ferroptosis are promising strategies for disease treatments such as cancer or neurodegenerative and cardiovascular diseases, respectively. Many drugs have been developed to exert ferroptosis-inducing and/or inhibiting reactions, such as erastin and iron-chelating compounds, respectively. In addition, many natural bioactive compounds have significantly contributed to regulating ferroptosis and ferroptosis-induced oxidative stress. Natural bioactive compounds are largely abundant in food and plants and have been for a long time, inspiring the development of various low-toxic therapeutic drugs. Currently, functional bioactive peptides are widely reported for their antioxidant properties and application in human disease treatment. The scientific evidence from biochemical and in vitro tests of these peptides strongly supports the existence of a relationship between their antioxidant properties (such as iron chelation) and ferroptosis regulation. In this review, we answer questions concerning ferroptosis milestones, its importance in physiopathology mechanisms, and its downstream regulatory mechanisms. We also address ferroptosis regulatory natural compounds as well as provide promising thoughts about bioactive peptides.

## 1. Introduction

Cell death is a physiological and pathophysiological process leading to human cell renewal. It is the final point of the cell cycle controlled by canonical mechanisms (apoptosis, necrosis) to more exotic ones such as efferocytosis, necroptosis, pyroptosis, parthanatos, and ferroptosis. Ferroptosis is an iron-dependent form of non-apoptotic cell death, characterized by lipid peroxidation due to glutathione depletion and a decrease of glutathione peroxidase 4 (GPx4) catalytic activity.

The mechanism leading to ferroptosis has evolved from a sequence of atmospheric and ecological events. Since the early days on earth, trace amounts of oxygen have been produced from photosynthesis. Then they became abundant after the Great Oxygenation Event [1], which was the starting point of free radicals forming inside eukaryotic cells [2]. Dioxygen (O_2_) is a strong oxidant that needs activation to be reactive. For example, O_2_ is the last acceptor of electrons and protons produced by the respiratory chain. Moreover, O_2_ drives the oxidation of iron, forming reactive oxygen species that, in turn, oxidize polyunsaturated fatty acids (PUFA) of the cell membrane. The abundant iron-dependent cell mechanisms made this oxidation phenomenon even faster [3]. Early cell studies suggested the evolution of defense mechanisms to counteract the iron/oxygen-driven lipid peroxidation or to cause cell death [4]. These defense mechanisms, referred to as antioxidant power, reside in the intracellular and extracellular environment under enzymatic and non-enzymatic forms.

Therefore, ferroptosis is an ancestral type of cell death discovered only in 2012 by Dixon et al. using pharmacological approaches [5]. Dixon et al. transpose this physiological mechanism to cancer therapy. The timeline of discoveries leading to the current definition of the ferroptosis mechanism is presented in Figure 1. Ferroptosis comes from the Latin word “Ferrum”, meaning “iron”, and the Greek word “ptosis”, which means “to fall” [6]. Between 2012 and 2022, the scientific interest in ferroptosis increased exponentially, with more than 5000 articles published, with about half published in 2022. Research concerning cancer therapy targeting ferroptosis has guided the development of many ferroptosis inducers [7,8]. According to PubMed statistics, 1264 articles concerning ferroptosis inducers were published; on the other hand, 522 articles about ferroptosis inhibitors are published in 2022 (www.pubmed.ncbi.nlm.nih.gov (accessed on 30 September 2022)). However, the implication and treatment of ferroptosis in several diseases is still an ill-defined subject.

Nature-derived antioxidants from various sources show great bioactivities with a broad application spectrum in the prevention and/or treatment of diseases related to oxidative stress. Their mechanisms of action are extensive and include notably the reduction/scavenging of reactive oxygen species (ROS), the inhibition of lipid peroxidation, and the chelation of free metal ions [9]. In addition, several studies demonstrated that many natural compounds could regulate ferroptosis and could be promising drugs for targeting ferroptosis-related pathologies. Among them, bioactive peptides are also considered antioxidant compounds. With their functional amino acid sequences, these peptides can react with all possible oxidants and inhibit or suppress their damaging activities [10]. However, the bioactive peptides and ferroptosis relationship are still unexplored avenues in the research.

**Figure 1 molecules-28-02636-f001:**
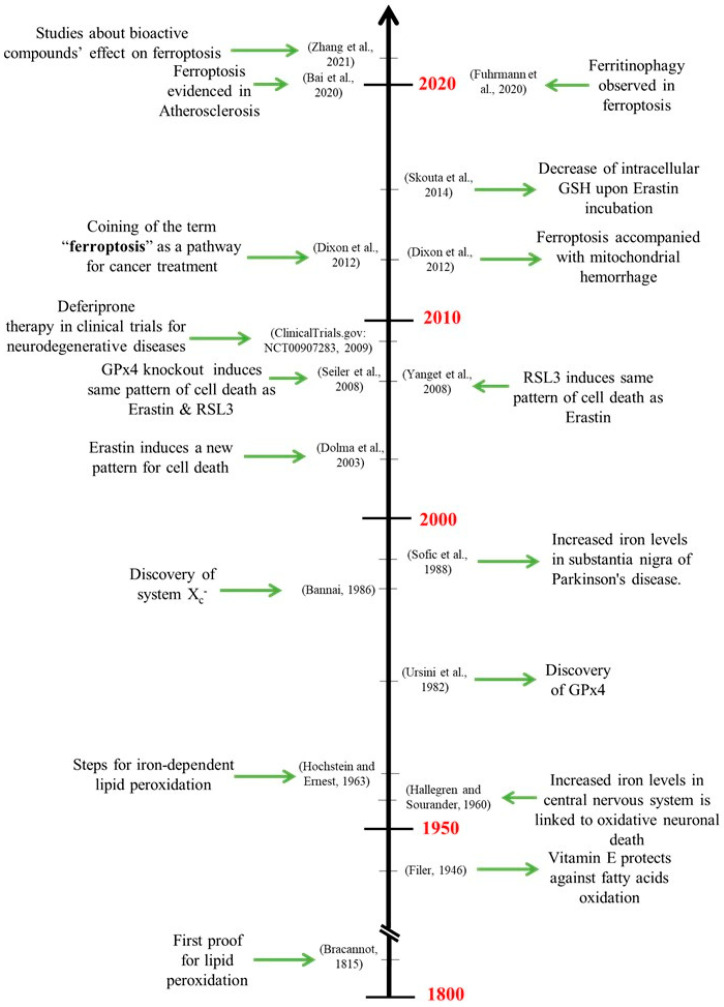
Timeline of the major discoveries concerning ferroptosis [5,11,12,13,14,15,16,17,18,19,20,21,22,23,24,25].

In this review, we highlight the main cellular and molecular mechanisms through which ferroptosis is induced and/or inhibited and their relationship to cardiovascular and neurodegenerative diseases. In addition, we state here the antioxidant properties of naturally derived compounds and bioactive peptides that have promising effects on ferroptosis-related conditions.

## 2. Iron-Catalyzed Lipid Peroxidation in Cells

Iron is an important trace element for cell functions [26]. However, its distribution and content should be controlled constantly in order to prevent iron overload or iron deficiency-related diseases [27]. Iron is absorbed by the intestines at the duodenum and proximal jejunum level, where it exists in its ferrous (Fe^2+^) form because of the low pH (∼5–6) in the duodenum, mainly due to gastric acid. When absorbed, Fe^2+^ is oxidized to Fe^3+^ by blood proteins such as hephaestin and ceruloplasmin. The majority of iron is transported and stored as a complex with proteins. Indeed, in the bloodstream, Fe^3+^ binds to transferrin, which transports Fe^3+^ from the bloodstream to tissues. The transferrin–Fe^3+^ complex is endocytosed after binding to transferrin receptors on the plasma membrane of target cells. The low pH of the endocytosis vesicle associated with ferrireductase activity reduces Fe^3+^ to Fe^2+^. Inside cells, Fe^2+^ is stored in ferritin and/or within the labile iron pool (LIP) [28,29,30]. Free iron (Fe^2+^ and Fe^3+^) provokes harmful reactions with proteins, nucleic acids, and lipids in the plasma membrane. Notably, iron excess participates in lipid peroxidation via hydroxyl-dependent and hydroxyl-independent mechanisms [31]. In the first mechanism, Fe^2+^ is oxidized into Fe^3+^ by hydrogen peroxide H_2_O_2_ through the Fenton reaction that produces hydroxyl radical OH^●^. In cells, Fe^3+^ produced by the Fenton reaction can be reduced back to Fe^2+^ by the oxidation of O_2_^●−^ to O_2,_ originating a catalytic cycle [32]. The formed hydroxyl radical is a highly reactive oxidant (half-life of 10^−9^ s) extracting hydrogen atoms from lipid molecules, starting lipid peroxidation [33]. The process of lipid peroxidation involves three main steps: initiation, propagation, and termination (Figure 2). For initiation, the lipid (L-H) is converted to a radical (L^●^) by a highly reactive species such as the hydroxyl radical, which pulls a hydrogen atom. Then, during the propagation step, L^●^ quickly reacts with dioxygen (O_2_) and thus forms a lipid peroxyl radical (LOO^●^), which in turn extracts hydrogen from another L-H to form lipid hydroperoxide (LOOH) and a new L^●^. This is the propagation step. Finally, the termination step happens when a high concentration of lipid radical species is reached, so the probability of a collision between them is high. When two radical lipid species interact (2L^●^, 2LOO^●^, L^●^ + LOO^●^), a non-radical species is formed [34,35]. Lipid peroxidation forms a wide range of reactive aldehydes, such as malondialdehyde (MDA) and 4-hydroxynonenal (HNE), that can be mutagenic and carcinogenic [36,37,38]. If lipid peroxidation is not rapidly terminated, damage to cell membranes takes place. Moreover, lipid peroxidation is implicated in the initiation and development of pathologies such as atherosclerosis. Atherosclerosis is a cardiovascular disease induced by an accumulation of oxidized lipids in the vascular wall forming atherosclerotic plaques. The early stage of plaque formation involves the oxidation of low-density lipoproteins (LDL), which attract macrophages. Those macrophages phagocyte oxidized LDL forming foam cells that compose atherosclerotic plaques [39,40,41].

## 3. Ferroptosis, a Cell Death Type among Others

The definition of cell death was first coined in 1858 by the biologist Rudolf Virchow, but the apoptosis mechanism (programmed cell death) was only elucidated in the 20th century. Besides canonical mechanisms of cell death (apoptosis, necrosis), more exotic ones such as efferocytosis, necroptosis, pyroptosis, parthanatos, and ferroptosis have been described. These types of cell death exhibit some similar features but remarkable differences in terms of morphology, mechanisms, and consequences [42].

Apoptosis is a programmed cell death mechanism that occurs for advantageous reasons for the organism’s life cycle [43]. It plays an important role during embryonic development as it shapes formed organs [44]. Apoptosis is a “self-suicide” that can be triggered by either intrinsic factors (i.e., cell aging, mutations, toxic infection...) or extrinsic ones (i.e., adjacent cells, cytokines…) activating caspase enzymes [45]. These enzymes degrade proteins and start the cellular execution process with specific morphologies characterized by DNA fragmentation, chromatin condensation, nuclear fragmentation, cell shrinkage, and blebbing [46]. Finally, cell fragments are released in the form of apoptotic bodies, then engulfed by phagocytosis cells via a mechanism called “efferocytosis”. Furthermore, deCathelineau and Henson nicely described efferocytosis as the process where phagocytes take apoptotic cells to their grave [47]. Necrosis, on the other hand, stands on the opposite pole from apoptosis. It is often a deadly process for tissues and organs initiated by an unregulated degradation of cell components and induced by extrinsic factors such as mechanical injuries, heat shocks, and pathological infections [48]. The fate of a necrotic cell is characterized by the rupture of its membrane and the release of its components into the extracellular space [49]. Necroptosis is a programmed form of necrosis, which is not always detrimental to biological systems. On the contrary, it contributes to defense mechanisms against viral infections. When cells are unable to undergo apoptosis due to the inactivation of apoptotic proteins by viral proteins, necroptosis is activated, independently from caspases, by responding to a pro-inflammatory signal from macrophages [50]. During necroptosis, like during necrosis, ATP levels drop, and the cell membrane is ruptured, leading to product leakages but in a regulated manner that is still under investigation [51]. Pyroptosis also falls in the category of programmed cell death. It generally takes place in immune cells during pathogenic infections. It is initiated by the cell itself when it starts to assemble the inflammasome, defined as an agglomeration of proteins that promotes multiple pro-inflammatory pathways [52]. Like in apoptosis, caspases are then activated; however, blebbing does not appear, but membrane pores do, and as a result, all molecules are released outside cells [53]. Parthanatos is a caspase-independent cell death with certain apoptotic features [54]. It is induced by extreme DNA damage, which is directly recognized by intracellular enzymes that activate the expression of the DNA repair factor, poly-ADP ribose (PAR). The more extreme the DNA damage is, the longer and more branched PAR is and the more toxic effect it has on the cell. PAR can also bind and activate the apoptosis-inducing factor (AIF), leading to its translocation from the mitochondria to the nucleus to start chromatin condensation and large-scale DNA fragmentation. Cells finally die by losing their membrane integrity [55].

Ferroptosis is a regulated form of cell death characterized by lipid peroxidation catalyzed by iron overload. Although studies concerning its biochemical and morphological features are still in their early stages, its distinction from other types of cell death was obvious from the beginning of its discovery. Ferroptosis can propagate to adjacent cells, which are swollen, rounded up, and detached from each other [5,56,57]. Nuclear integrity is maintained, but chromatin condensation is missing during ferroptosis [58]. Mitochondrial alterations are the most obvious changes observed during ferroptosis. The mitochondria appear smaller with thicker membrane density and reduced crista that will eventually lead to its destruction [5,18,59,60]. In physiological conditions, mitochondrial iron content is regulated by iron exporters and mitochondrial ferritin. However, the mitochondrial iron content is very high in certain ferroptosis-related diseases or ferroptosis-induced cancer cells [61,62,63,64]. Iron overload is accompanied by an increase in mitochondrial ROS production and reduced mitochondrial glutathione (GSH) content [62,65,66]. Decreased mitochondrial potential and increased mitochondrial permeability are other ferroptosis signs [59,67]. Recently, Guo et al. showed that inhibiting the replication of mitochondrial DNA can induce ferroptosis [68]. Moreover, internal lysosomal autophagy of ferritin, so-called “ferritinophagy”, is also observed in ferroptotic cells [13,59]. Ferroptosis is similar to H_2_O_2_-induced necrosis by the rupture of the plasma membrane but, yet, is not accompanied by ATP depletion [5,69,70]. All in all, ferroptotic cells accumulate iron, lipid peroxides, oxidized NADPH, and glutamate. They are depleted in cystine/cysteine and GSH [60,71,72].

## 4. Physiological Regulation of Ferroptosis

For properly functioning cells, biomolecular regulations are fundamental processes that adapt to internal and external environments. For example, cell oxidation caused by aerobic respiration is a physiologic phenomenon that maintains cell redox homeostasis. Impairments in the homeostatic control between cell oxidation and reduction reactions cause oxidative stress, eventually leading to lipid peroxidation up to ferroptosis (Figure 3).

The regulation of intracellular iron content is an important factor for resisting ferroptosis. Iron content is regulated at gene, cytosolic, and plasma membrane levels. Indeed, the *Iron Responsive element Protein 2* (*IREBP2*) gene encodes for the iron-responsive element binding protein (IRE-BP) that binds to iron-responsive elements (stem-loop RNA structures) of many genes such as *ferritin heavy chain 1* (*FTH1*) and *transferrin receptor* (*TFRC*) to control cell iron homeostasis. Therefore, variants in the *IREBP2* gene affect the physiological cell iron metabolism and can aid in ferroptosis [5]. Furthermore, in the cytosol, ferritinophagy is caused by the upregulation of autophagy-related proteins (ATG5 and ARC7) that release the ferritin-stored iron and, thus, induce ferroptosis [73]. Finally, at the plasma membrane level, a too-high activity of transferrin and transferrin receptors mediates ferroptosis by iron overload inside cells [74].

The hallmark of ferroptosis might also be modulated by enzymes implicated in lipid peroxidation processes. Indeed, several cells knockout for Acyl-CoA synthase long-chain family 4 (ACSL4) and lysophosphatidylcholine acyltransferase 3 (LPCAT3), implicated in the oxidation of membranes constituted of poly-unsaturated fatty acids [75,76], and are resistant to ferroptosis. In addition, lipoxygenase (LOX) is a ferroptosis-promoting enzyme that oxidizes phosphatidylethanolamine found in cell membranes [77]. Indeed, LOX overexpression in human kidney cells results in their sensitization to ferroptosis [77]. Finally, Glutathione peroxidase 4 (GPx4), an antioxidant enzyme reducing peroxidized lipids, can prevent ferroptosis [78]. Indeed, among the eight human GPxs’, GPx4 is the only one that can reduce oxidized phospholipids present in membranes. Thus, its catalytic activity is critically important for cell health, and the conditional deficiency of GPx4 in cells causes ferroptosis [78]. GPx4 is a selenoprotein that requires selenocysteine for its antioxidant activity [79]. Yet, the synthesis of selenoproteins requires isopentenyl-PP, an intermediate by-product of the mevalonate pathway [80]. Moreover, the mevalonate pathway produces, in addition, the coenzyme Q_10_, which has an antioxidant function at the membrane level [72]. Thus, the depletion of selenium provokes ferroptosis due to GPx4 misfunctioning [81]. Moreover, the activity of GPx4 strongly relies on the content of glutathione (GSH), its cofactor, inside cells [21].

The synthesis of intracellular GSH requires cysteine [82], which can be either provided via the transsulfuration pathway where methionine is converted to S-adenosylhomocysteine and cysteine [71] or via transmembrane proteins responsible for intracellular cysteine transport such as the L-amino acid transporter (L-AT) or system X_c_^−^. System X_c_^−^ is a glutamate/cystine antiporter critical for providing cystine to cells. Cystine is then reduced to cysteine required for GSH synthesis.

In summary, ferroptosis is a cell death type related to intrinsic cellular events that can be modulated by the cell’s autonomous properties, including iron metabolism and redox reactions. However, the discovery and understanding of ferroptosis are being initiated in several studies through the action of pharmacological ferroptosis inducers and inhibitors.

## 5. Pharmacological Modulation of Ferroptosis

### 5.1. Ferroptosis Inducers

Ferroptosis, as previously clarified, was first discovered as a novel therapeutic approach to killing cancer cells using pharmacological agents [5]. Nowadays, a variety of molecules are used to induce ferroptosis, whether to treat cancer or to clarify cell pathways implicated in ferroptosis induction and inhibition (Figure 4). Ferroptosis inducers are reviewed in several publications [69,83,84]. The two most popular targets for ferroptosis inducers are system X_c_^−^ and GPx4 (Figure 5). Inhibitors of system X_c_^−^ such as erastin, sulfasalazine, and sorafenib are class 1 ferroptosis inducing compounds (FIN I). In contrast, inhibitors of GPx4, such as Ras selective lethal 3 (RSL3), are class 2 ferroptosis inducing compounds (FIN II).

Blocking system X_c_^−^ is a well-known mechanism to provoke ferroptosis [66]. To that purpose, different strategies are reported in the literature, such as glutamate excess, the glutaminolysis pathway [85], and system X_c_^−^ direct inhibition by erastin. Erastin is a well-known inhibitor of system X_c_^−^ decreasing the intracellular concentration of cystine [86]. Erastin-induced ferroptosis can also promote endoplasmic reticulum stress [56]. Erastin was initially described as a cancer cell killer via a pathway different from apoptosis [18], which was more recently identified as ferroptosis [5]. Indeed, erastin has been shown to enhance cell sensitivity to chemotherapy in lung cancer, rhabdomyosarcoma, and glioblastoma models [86,87,88] and to alleviate cell resistance to chemotherapeutic drugs in head, neck, and ovarian cancer cells [89,90]. Erastin is also a radiosensitizer in radiotherapy [91,92,93]. However, erastin is rarely tested in vivo because of its poor water solubility. Piperazine-erastin (PE) and imidazole ketone erastin (IKE) are water-soluble erastin-based compounds. PE was shown to decrease tumor cell growth in mice, and IKE induced ferroptosis in a B cell lymphoma model [94,95]. Sulfasalazine and sorafenib are pharmacological reagents that induce ferroptosis in several cancer types by inactivating system X_c_^−^ [66,96,97,98,99,100,101]. However, erastin has a higher inhibitory effect on system X_c_^−^ than the two other molecules, sulfasalazine and sorafenib [66]. Other strategies, such as engineered “cyst(e)inase”, which degrades cell cystine required for GSH production, showed similar results on lymphatic leukemia with the system X_c_^−^ blocking strategy [102]. GSH synthesis can also be inhibited by buthionine sulfoximine by inactivating 𝛾-glutamylcysteine synthase.

The second target to induce ferroptosis is GPx4. RSL3, a FIN II, reduces GPx4 catalytic activity and thus drives cells to ferroptosis [103] in colorectal cancer, for example [104]. On the contrary, RSL3-driven ferroptosis can be reversed by GPx4 overexpression [96]. Withaferin A and Altretamine also belong to FIN II and induce ferroptosis in neuroblastoma and ovarian cells, respectively [105,106]. Other compounds, such as ML-210 (DPI10) and ML-162 (DPI7), are also classified as FIN II [107,108]. FIN56 is a ferroptosis-inducer classified in class 3 ferroptosis inducers, although it promotes the degradation of GPx4 through a not yet fully described mechanism [109] that is potentiated by statins [72]. FIN56 also binds and activates squalene synthase leading to the depletion of the lipophilic antioxidant coenzyme Q10. Finally, FINO_2_, an endoperoxide-containing 1,2-dioxolane, indirectly inactivates GPx4 through a yet unknown pathway. Moreover, FINO_2_ can react with Fe^2+^ to produce radicals and aid in lipid peroxidation [110]. Besides inhibitors of X_c_^−^ and of GPx4, ferroptosis can also be initiated by increasing the labile iron pool (LIP) through the excessive expression of transferrin, limited expression of ferritin, or activation of heme-oxygenase [105,111,112].

### 5.2. Ferroptosis Inhibitors

Following the research around ferroptosis mechanisms, specific ferroptosis inhibitors such as metal-chelators such as deferoxamine (DFO) and deferiprone, and lipophilic antioxidants such as ferrostatin-1 (Fer-1), liproxstatin-1 (Lip-1), and vitamin E (⍺-tocopherol) have been discovered (Figure 5).

DFO (Figure 6A) is a metal-chelating agent for trivalent cations such as ferric and aluminum ions. The affinity constants of complex formation are very high, 10^31^ M^−1^ and 10^25^ M^−1^, respectively, and are carried out on a 1:1 molar basis. DFO is able to bind free iron in plasma or cells to form the ferrioxamine complex. It can also chelate iron bound to ferritin but cannot chelate iron in hemoglobin. As a result, DFO promotes the excretion of iron and aluminum in urine and feces, thereby reducing their pathological effects in organs. Interestingly, iron and aluminum overload can be screened by the DFO test. This test is based on the fact that DFO does not increase the excretion of iron and aluminum above a certain threshold in healthy subjects. DFO is prescribed for patients suffering from primary hemochromatosis (extra iron builds up in the body to harmful levels caused by gene disorders), which is not curable by phlebotomy, secondary hemochromatosis (extra iron builds up in the body to harmful levels caused by another disease), and renal aluminum poisoning. It can be administered either subcutaneously, intravenously by infusion, intramuscularly, or intraperitoneally. DFO has shown interesting results in ferroptosis-based studies. For example, spinal cord injury in mice was treated by DFO decreased iron concentration and promoted the expression of GPx4, system X_c_^−^, and GSH [113]. DFO is also reported to preserve neurons in the early stages of Parkinson’s disease [114] and to reduce ROS formation in breast cancer cells and corneal epithelial cells driven to ferroptosis [115,116]. Deferiprone (Figure 6B) is a bidentate ligand that binds to iron in a molar ratio of 3:1.

Unlike DFO, deferiprone is a membrane-permeant molecule. Deferiprone promotes iron excretion in transfusion-dependent β-thalassemia. This drug is administered orally and prescribed as a monotherapy to treat iron overload when current chelation therapy is unsuitable or in combination with another chelating drug (mainly DFO) when the monotherapy is ineffective or in the case of life-threatening iron overload. To date, deferiprone has never been tested on a ferroptosis model, although it presents similar properties as DFO. For example, it was shown to reduce intracellular iron levels and inhibit lipid peroxidation in hepatocytes [117]. Deferiprone was also able to chelate iron deposits in the red blood cells of patients with thalassemia and alleviate plasma membrane dysfunction.

Ferrostatin 1 (Fer-1; Figure 6C) is a ferroptosis inhibitor that prevents lipid peroxidation by scavenging lipid alkoxyl radicals [118]. Dixon et al. were the first to identify its rescuing effect on glutamate-induced ferroptosis in rat brains [5]. This molecule was also able to restore mitochondrial function in ferroptosis-based neuronal cell models and prevent their death [119]. Furthermore, Fer-1 showed antioxidant effects in cell models of Huntington’s disease, kidney dysfunction, atherosclerosis, and lung cancer by reducing oxidation of lipids and ROS production, and it indirectly promoted the expression of GPx4 [12,14,120]. These later findings suggest that ferroptosis is a key event in the progression of these diseases. Lipophilic antioxidants such as Lip-1 and vitamin E also showed anti-ferroptosis effects. To date, the Lip-1 mode of action is not well explained in the literature, but it is believed to have the same scavenging activity as Fer-1 due to the presence of two amine groups in its structure (Figure 6D) [121]. Furthermore, Lip-1 was able to inhibit ferroptosis in mice fibroblasts where GPx4 was knocked-out [58]. Vitamin E—the lipid-soluble form of ⍺-tocopherol that blocks oxidation chain reaction in the membrane—is less efficient than Fer-1 and Lip-1 [6,122]. Vitamin E suppressed FIN56-induced ferroptosis in fibrosarcoma cells and prevented lipid peroxidation-induced cell death in acute liver failure [110,123].

Ferroptosis inducers and inhibitors have represented a breakthrough in understanding this cell-death mechanism. They have also set up the basis to discover other ferroptosis regulatory molecules.

## 6. Ferroptosis in Cardiovascular and Neurodegenerative Diseases

Atherosclerosis is a cardiovascular disease characterized by the formation of plaques at the artery’s wall, narrowing the blood flow passage. In the context of inflammation and oxidative stress, atherosclerosis starts with an endothelial dysfunction associated with low-density lipoproteins (LDL) entry and oxidation in the sub-endothelium. Macrophages at that site phagocyte oxidized LDL and became foam cells appearing as a fatty streak [118]. Although the entire mechanism leading to atherosclerosis has not yet been clarified, several studies have provided a reliable correlation between ferroptosis markers and atherosclerosis (Figure 7). Oxidized phospholipids and iron overload are master players in ferroptosis [124] and are well-known for their role in the context of oxidative stress leading to atherosclerosis [119,125,126]. Foam cells of atherosclerotic plaques are found to be overloaded in transferrin-receptor proteins, one of the endogenous ferroptosis inducers [127]. Moreover, Vinchi et al., showed that non-transferrin-bound iron had provoked pro-atherosclerotic responses in mice, such as endothelial permeabilization and activation, elevated oxidized LDL, macrophage recruitment, plaque formation, and death of vascular cells [128]. Iron overload and lipid peroxidation in macrophages has been attributed in many studies to their accumulation, pro-inflammatory response, and death in the atherosclerosis plaque [129,130,131,132]. In the same context, LOX overexpression—a cell ferroptosis inducer—also leads to atherosclerosis [124]. In a recent study, Sampilvanjil et al. induced ferroptosis in vascular smooth muscle cells with tobacco smoking [133], known as a key factor for the development of atherosclerosis [134]. GPx4, a ferroptosis-resistant enzyme, plays a major role in the development of atherosclerosis. Overexpression of Gpx4 reduced atherosclerotic lesions in mice, as discussed by Guo et al. [135]. Early studies showed that selenium supplementation, a co-factor for GPx4 activity, can stop the development of atherosclerosis plaques and reduce lesions in animal models [136,137,138]. Furthermore, ferroptosis inhibitors, such as Fer-1, DFO, and vitamin E, can prevent atherosclerosis [139,140]. Indeed, Bai et al. showed that Fer-1 inhibits lipid peroxidation and iron accumulation and induces GPx4 overexpression as well as GSH production in an atherosclerosis mice model [12]. DFO, on the other hand, has been shown to reduce atherosclerotic inflammation and lesions, even before the discovery of ferroptosis [141]. Wang and Tang inhibited endothelial cell ferroptosis using Fer-1 and DFO [142]. Other cardiovascular diseases, such as cardiomyopathy and cardiac dysfunction, are also induced by iron overload [143,144]. Indeed, cardiomyopathy appears in cancer patients treated with doxorubicin due to its ability to accumulate iron and thus induce ferroptosis [145]. According to some studies, ferroptosis is a leading cause of myocardial I/R injury (tissue damage caused by the return of blood supply after a period of lack of oxygen) whose damage can be reduced by Fer-1 and DFO [85,146,147]. Lip-1 was able to regulate mitochondrial ROS and restore GPx4 levels in the heart of the I/R injury mouse model [148]. Fer-1 was shown to inhibit cell death and restore redox balance in a cell model of ischemic heart disease induced by H_2_O_2_ [149]. A study suggested that prior treatment with Fer-1 before heart transplantation surgeries would avoid graft dysfunctions [146]. Other cardiovascular conditions, such as heart failure, vascular injury, and strokes, were reviewed with their ferroptosis biomarkers [150].

Several scientific data support the relationship between ferroptosis and other chronic diseases, such as Huntington’s, a genetic neurodegenerative disease. Biomarkers of this disease, including excessive iron metabolism, glutamate toxicity, dysregulation in GSH-mediated redox potential and lipid peroxidation, are also markers of ferroptosis [151,152,153]. DFO and Fer-1 protected the death of neurons in Huntington models [14,154]. In addition, Alzheimer’s disease and Parkinson’s disease are also common neurodegenerative diseases associated with loss of memory and automaticity, respectively. Several features of these diseases’ physiopathology are consistent with ferroptosis pathways (Figure 7). Long before the term ferroptosis appeared, scientists observed increased iron levels in the substantia nigra region of Parkinson’s disease brains [19,155,156,157], causing elevated lipid peroxides and oxygen free radicals [158,159,160,161]. The loss of nigral cells due to oxidative damage is a major characteristic of this disease’s severity [162]. In Alzheimer’s disease, alterations in iron levels and iron regulatory proteins (ferritin and transferrin) associated with lipid peroxidation and oxidative stress were observed in the hippocampus [23,163,164,165,166,167,168]. Nowadays, ferroptosis has been introduced as a key player in the progression of many neurodegenerative diseases [169,170,171]. Low expressions of antioxidant machinery proteins, including GPx4, GSH, system X_c_^−^, and CoQ_10_, and mitochondrial dysfunctions, were also reported in the degenerative neurons [172,173,174,175,176,177]. Ferritinophagy is also a ferroptosis process implicated in specific neurotoxicities [178]. In addition, antioxidant therapies using ferroptosis inhibitors (Fer-1, Lip-1, DFr, and DFO) and targeting pathological properties of neurodegenerative diseases have achieved promising results on cellular and animal models [14,114,179,180]. Dietary supplementations with the antioxidant CoQ_10_ have also been discussed as a potential neuroprotective candidate [181]. 

Interestingly, iron chelation therapy for neurodegenerative diseases has recently been featured in several clinical trials. Early-stage Parkinson’s disease patients were administered different doses of DFr for 6 months in an independent phase 2 double-blind, placebo-controlled randomized trial [15]. As a blood–brain barrier permeable iron chelator, DFr was well tolerated by the patients in this trial but had a non-significant trend to reduce iron content and improve quality of life [182]. Similarly, DFr was tested in a large phase 3, European multicentric, parallel-group, placebo-controlled randomized trial for Parkinson’s disease patients, but results are not yet published [183,184]. This iron-chelating drug was also evaluated in a multicenter, unblinded, single-arm pilot trial on patients with cerebral iron overloads (clinical symptoms: strong disability and reduction of autonomy) [185]. This trial documented a reduction in iron accumulations and improvements in the motor activities of patients [186]. In France, there is now a promising, still active, double-blind, placebo-controlled randomized study for DFr on patients with amyotrophic lateral sclerosis (a neurodegenerative disease that affects the brain and spinal cord) and causes loss of muscle control [187].

The discovery of ferroptosis has led to a better understanding of various diseases, and even though it is not yet mentioned in clinical studies, its inducers and inhibitors highlight a promising future for disease treatments.

## 7. Ferroptosis Modulation Inspired by Nature

### 7.1. Natural Compounds

For many years, the drug development industry has relied on natural products to treat and prevent various diseases. Bioactive molecules extracted from natural resources have been proven to possess anti-cancer, anti-inflammatory, antibacterial, immunomodulatory, and antioxidant activities. However, to date, only a few published studies directly correlate the effect of natural compounds on the regulation of iron metabolism, lipid peroxidation, GPx4 activity, and other pathways linked to ferroptosis induction or inhibition.

In 2015, Tu Youyou shared the Nobel Prize in Medicine for her discovery of artemisinin from the plant *Artemisia annua* as an antimalaria drug, which turned out to be a promising drug for various other diseases, including cancer. Further studies showed that artemisinin and its derivatives could stimulate ferroptosis cell pathways, including the downregulation of GPx4 expression, increasing the production of ROS and intracellular iron. Other natural-based compounds have been defined as ferroptosis inducers, such as Ferroptocide and Erianin. Ferroptocide, a stereochemical derivative of a fungi metabolite called “Pleuromutilin”, can induce ferroptosis in breast cancer cells by inactivating the thioredoxin, a cell antioxidant enzyme [188]. Chen et al. demonstrated that Erianin, a natural product isolated from *Dendrobium chrysotoxum Lindl*, induced ferroptosis in lung cancer cells where ROS accumulation, lipid peroxidation, and GSH depletion were observed [189].

Certain bioactive natural compounds, such as Ruscogenin and Curcumin, may exert ferroptosis regulatory activities. While these compounds were shown to induce ferroptosis for cancer treatments, they can also act as ferroptosis inhibitors for tissue injury treatments. However, Ruscogenin is an important steroid sapogenin derived from *Ophiopogon japonicus.* In pancreatic cancer cells, it increased intracellular ferrous ions as well as ROS production [190]. While in myocardial injury and acute kidney injury mice models, this compound has led to the upregulation of GPx4 expression, an increase in GSH concentration, and a decrease of intracellular iron and lipid peroxidation in a dose-dependent manner [191,192]. Curcumin from *Curcuma longa* has stimulated cell death by ferroptosis via accumulating iron ions in breast cancer cells and sabotaging GPx4 activity in glioblastoma cells [193,194]. On the contrary, Curcumin triggered the antioxidant defense mechanism against induced ferroptosis in renal tubular cells and thus decreased lipid peroxidation [195]. This controversiality requires more studies that underline the biochemical and/or cellular conditions that regulate these compounds’ mode of action.

Treating ferroptosis-induced diseases with natural bioactive compounds has also been investigated. Flavonoids such as Baicalein, Kaempferol, and Quercetin have long been known for their antioxidant properties in cardiovascular and neurodegenerative diseases. Baicalein, derived from the root of *Scutellaria baicalensis*, has been served as a food supplement in many countries [196]. In 2009, Perez and his colleagues demonstrated that this compound protects against the cellular Fenton reaction by chelating ferrous ions [197]. In a recent study in a posttraumatic epilepsy rat model, Baicalein revealed its neuroprotective effects by decreasing the production of lipid peroxides [198]. Kaempferol is found in diverse foods, including broccoli, peaches, spinach, and green tea [199]. Yuan et al. showed that this antioxidant compound can deattenuate neuronal ferroptosis by enhancing the X_c_^−^/GPx4 signaling pathway, thus reducing lipid peroxidation [200]. Quercetin is also consumed via a wide variety of fruits, vegetables, and beans. It inhibited erastin-induced ferroptosis in bone marrow-derived mesenchymal stem cells via several antioxidant pathways, including lipid peroxide scavenging [201].

Although this review highlights a few examples of natural compounds that regulate ferroptosis, dozens of others are reported in the literature. Zhang et al., and Zheng et al., nicely reviewed the ferroptosis regulatory activity of natural compounds [11,202].

### 7.2. Bioactive Peptides

Bioactive peptides are short sequences (2 to 40 units) of characteristic amino acids initially encrypted inside the 3-dimensional structure of proteins. Although these peptides are widely reported for their antioxidant activities, such as radical scavenging, reducing power, and metal chelation, to date, there is no study investigating them in a ferroptosis model. Therefore, in this part, we will review the antioxidant properties of bioactive peptides such as metal chelation, reduction of lipid peroxidation, modulation of GSH content, and GPx activity that might confer their interest for anti-ferroptotic properties. These peptides can be found, synthesized, and/or produced by bacteria, plants, animals, and humans. For example, carnosine is a bioactive dipeptide produced in the liver from beta-alanine and histidine and has ROS-scavenging power, a known anti-ferroptosis pathway [203]. 

Moreover, GSH, the important ferroptosis cell regulator, is a bioactive tripeptide with a variety of antioxidant properties thanks to its special amino acid sequence (γ-l-glutamyl-l-cysteinylglycine) [82]. Since the antioxidant defense mechanism of human beings is not enough to fight against all the oxidative stress that we encounter in our lives, the discovery and extraction of external bioactive peptides are essential. These peptides can be produced in a variety of ways, including enzymatic hydrolysis of proteins, microbial fermentation, and chemical synthesis. Depending on the protein source, the production technique used, the processing condition, and the post-production treatments, peptide sequences, and their bioactive properties can vary. Nowadays, researchers race towards better-producing antioxidant peptides, most of which have promising inhibitory effects on ferroptosis.

Biochemical tests of bioactive peptides (whether in a mixture like peptide hydrolysates or pure) are the most common evaluation procedures used among researchers due to their easy operation, high efficiency, low toxicity, good reproducibility, and short experimental period. Evaluation methods based on metal ion chelating activity and radical scavenging power can be indicators for peptide anti-ferroptosis activity. Examples of these methods are DPPH and ABTS radical scavenging, the FRAP method, and the ORAC assay. In addition, the inhibition of lipid peroxidation has been screened using the TBARS method on a liposome peroxidation model [204]. Researchers have widely explored antioxidant peptides from different plant and animal origins (Table 1). The presence of amino acids, such as tyrosine (Y), tryptophan (W), methionine (M), lysine (K), proline (P), cystine (C), histidine (H), valine (V), leucine (L), and alanine (A), would be ascribed to different antioxidant activities possessed by these peptides [205]. For example, aromatic amino acids can donate protons to electron-deficient radicals, thus improving the radical scavenging capacity of peptides; also, histidine can donate hydrogen atoms, tap lipid peroxyl radicals, and/or chelate metal ions via its imidazole group, which traps lipid peroxyl radicals. The position of the amino acids inside the primary peptide sequence also plays a major role in the overall antioxidant capacity. Thus, histidine at the C terminus acts mainly as a scavenger of various radicals and at the N terminus as a metal ion chelator [206]. Chemical synthesis of metal-chelating and ROS-scavenging peptides have succeeded as well. For example, ref. [207] showed that the synthesized peptide sequences FDDDK and FEEEK acquire both iron (III)-chelating activity and HO. radical scavenging activity by spectrophotometric assays.

Peptides can also be evaluated via in vitro biological assays. Various cell-based oxidative stress models have been employed to investigate the antioxidant activity of peptides obtained from natural sources. Evaluation of these peptides mainly occurs by following up biomarkers such as cell viability, GSH, ROS production, lipid peroxidation, and antioxidant enzymes (SOD, CAT, and GPx) activities which are, as recently defined, biomarkers for ferroptosis regulation. The IYVVDLR, IYVFVR, VVFVDRL, and VIYVVDLR peptides from soybean hydrolysates were able to reduce lipid peroxidation and ROS production as well as increase cell viability and GSH content in H_2_O_2_-treated Caco2 cells [226]. In H_2_O_2_-treated HepG2 cells, the RDPEER peptide from watermelon seed protein hydrolysates and EDIVCW, MEPVW, and YWDAW peptides from Monkfish hydrolysates inhibited the production of lipid peroxides and other cells ROS [227,228]. Wang et al. determined that peptides DSTRTQ, DVYSF, and ESKPV can reduce ROS content in vascular smooth muscle cells [229]. The PHP peptide from Chinese Baijiu has decreased ROS and lipid peroxide content and elevated GSH concentration and antioxidant enzyme activities in 2,2′-azobis (2-methyl-propanimidamidine) dihydrochloride (AAPH)-treated HepG2 cells [230]. A more scientific basis is still required to determine the exact mechanisms of these peptides at the cell level.

As for in vivo evaluations, very few studies have reported the antioxidant activity of bioactive peptides on animal models. In diabetic mice, Jiang et al. identified that antioxidant peptides from red deer antlers could reduce malondialdehyde levels [231]. Ben Khaled et al. showed that sardine protein hydrolysates were able to reduce lipid peroxidation and enhance the activity of SOD and GPx in rats fed with a cholesterol-enriched diet [232]. Wheat bran protein hydrolysate had oxygen radical absorbance capacity in hypersensitive rats [233]. Finally, Ding et al. observed lower serum MDA levels in mice treated with jellyfish protein hydrolysates compared to non-treated mice [234]. Although experimenting on animal models is time-consuming and very sensitive, it provides a closer insight into the functionality of the organism. Yet tremendous scientific efforts are still required to determine the bioavailability, toxicity, required dose, and side effects of the various antioxidant peptides known to date.

## 8. Conclusions

In summary, the discovery of ferroptosis was a huge scientific progress in the biological domain. First, mainly used as a tool to kill cancer cells, ferroptosis was then discovered as implicated in cardiovascular and neurodegenerative diseases. Finally, the great value of identifying ferroptosis inducers and inhibitors was translated into disease-treatment studies. However, to date, ferroptosis has not been reported in any clinical trials (www.clinicaltrials.gov (accessed on 25 January 2022)). Therefore, more in-depth studies are required to make connections between cell/animal-based data and the complexity of the human organism. The main problem might be the lack of cell-based ferroptosis models related to those diseases to evaluate ferroptosis inhibitors. Nevertheless, scientists have also addressed people’s need to consume natural products by studying multiple different natural compounds on ferroptosis regulation. Although bioactive peptides have not been directly linked to ferroptosis regulation, their acquired various antioxidant activities, such as iron chelation properties, modulation of ferroptosis biomarkers, and restoration of redox homeostasis, might open a new area for the evaluation of those natural sources of ferroptosis-related conditions.

## Figures and Tables

**Figure 2 molecules-28-02636-f002:**
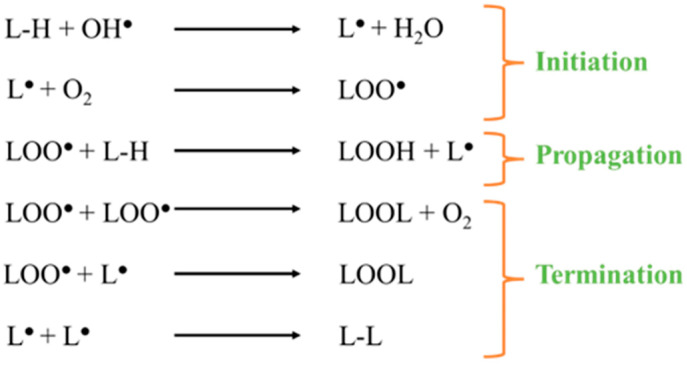
Process of lipid peroxidation.

**Figure 3 molecules-28-02636-f003:**
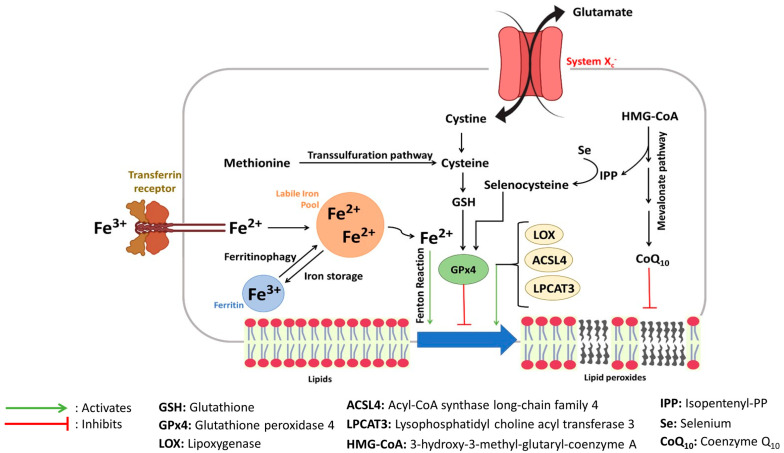
Network of ferroptosis signaling pathways.

**Figure 4 molecules-28-02636-f004:**
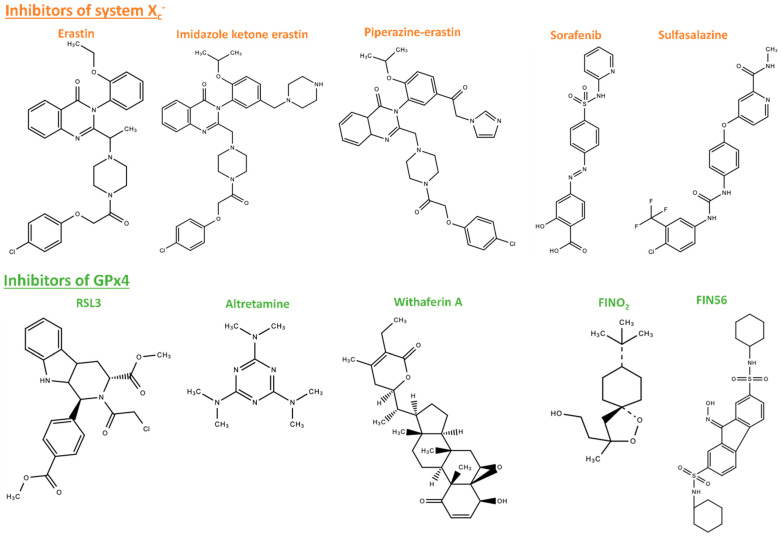
Chemical structures of ferroptosis inducers.

**Figure 5 molecules-28-02636-f005:**
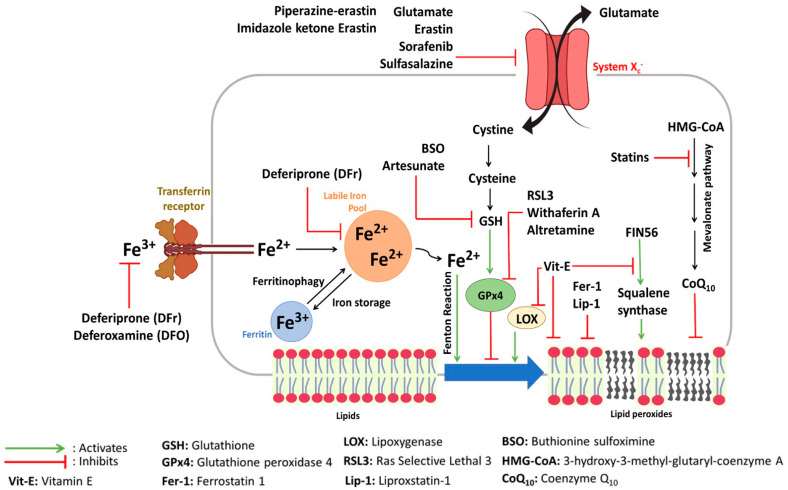
Network of ferroptosis inducers and inhibitors.

**Figure 6 molecules-28-02636-f006:**
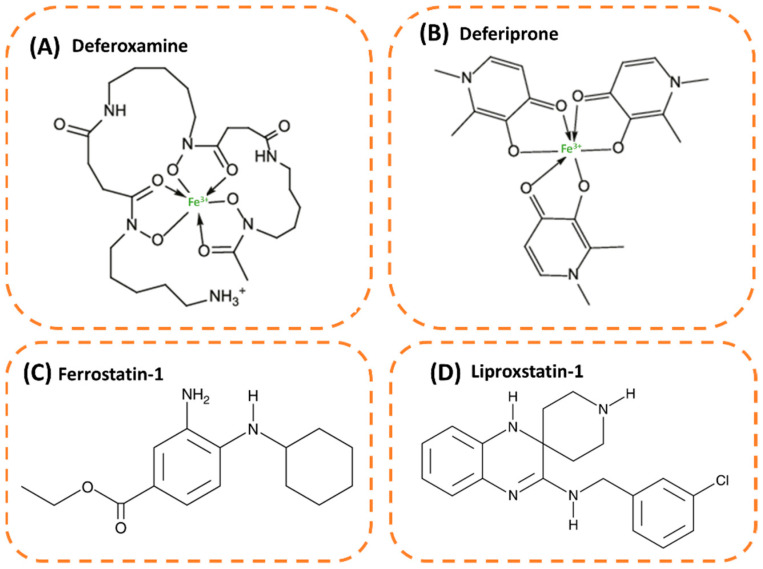
Chemical structures of iron chelators: (**A**) Deferoxamine chelating iron and (**B**) 3 Deferiprone chelating iron, and lipophilic antioxidants: (**C**) Ferrostatin-1 and (**D**) Lipoxstatin-1.

**Figure 7 molecules-28-02636-f007:**
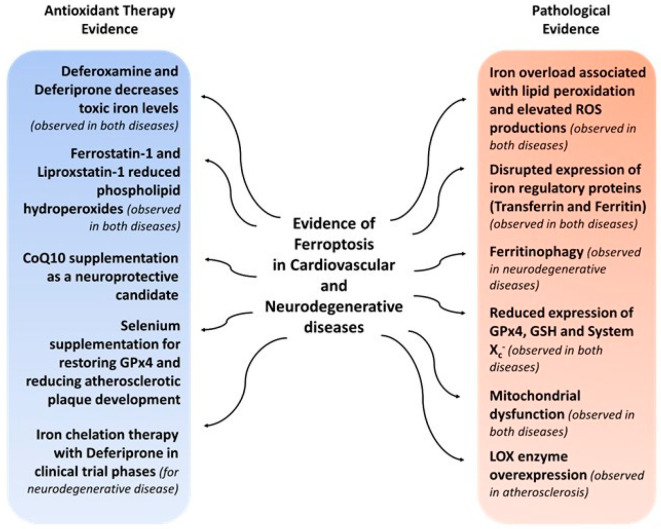
Ferroptosis evidence in Cardiovascular and Neurodegenerative diseases.

**Table 1 molecules-28-02636-t001:** Antioxidant peptide sequences from various plant and animal origins.

Origin	Peptide Sequences	Antioxidant Activities	References
Soy proteins	VNPESQQGSPR; LLPHH; LLPLPVLK; WLR; MLPVMR; SWLRL; SGDAL; SHECN; LPFAM	ABTS, DPPH, and Hydroxyl radical scavenging activities; Ferric reducing power; Inhibition of Fenton reaction	[208,209,210,211,212]
Rice proteins	FRDEHKK; RPNYTDA; TSQLLSDQ; TRTGDPFF; NFHPQ	ABTS, DPPH, and Hydroxyl radical scavenging activities; Lipid peroxidation inhibition in liposome model; Ferric reducing power	[141,208]
Cereal proteins (wheat, corn, pearl millet, barley)	KVALMSAGSMH; AGLPM; HALGA; SDRDLLGPNNQYLP; SVNVPL	Oxygen radical absorbance capacity; DPPH and Hydroxyl radical scavenging activities, Lipid peroxidation inhibition in liposome model; Iron (II) chelating activity	[206,209,210,211]
Seed proteins (rapeseed, walnuts, pine nut)	PAGPF; ADAF; KWFCT; QWFCT	DPPH radical scavenging activities; Lipid peroxidation inhibition in liposome model; Iron (II) chelating activity, Ferric reducing power	[212,213,214]
Egg proteins	WNIP; GWNI; WYGPD; KLSDW; KGLWE; IRW	Oxygen radical absorbance capacity; DPPH radical scavenging activity; Lipid peroxidation inhibition in liposome model	[215,216,217]
Fish proteins	VPKNYFHDIV; LVMFLDNQHRVIRH; FVNQPYLLYSVHMK; SCH; GVSGLHID; MTTL; LEW; YYPYQL; LEW; MTTL; YYPYQL	Oxygen radical absorbance capacity; ABTS and DDPH radical scavenging activities; Iron (II) chelating activity	[218,219,220,221]
Beef proteins	SNAAC; SAGNPN; DLEE; FWIIE; APYMM; GKFNV	Oxygen radical absorbance capacity; ABTS, DDPH, Hydroxyl, and Superoxide anion radical scavenging activities; Lipid peroxidation inhibition in liposome model; inhibition of Fenton’s reagent-induced protein oxidation; Iron (II) chelating activity	[222,223,224,225]

## Data Availability

No new data were created.

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
