# Peer review of "Nature-Inspired Bioactive Compounds: A Promising Approach for Ferroptosis-Linked Human Diseases?"

_molecules, 2023, doi:10.3390/molecules28062636_

Round 1
Reviewer 1 Report
The review focuses on ferroptosis, a cascade of biological response in cells overloading with iron ions. Iron belongs to trace elements present in animal cells, as necessary hem component. Iron ion Fe(II) is responsible for intracellular formation of ROS (different free radicals). ROS initiate reactions which have provided to cell death by random oxidation of membrane lipids or by destruction of nucleic acids. On the other hand, ferroptosis induction can be consider as therapeutic strategy in cancer treatment. The review contains description of different types of cell death including ferroptosis explanation of cellular regulatory mechanism responsible for homeostasis inside cells in case of ferroptosis induction. Additionally, the initiators and inhibitors of ferroptosis available on market and influence of ferroptosis in cardiovascular and neurodegenerative diseases is present. In the last part natural compounds and small peptides isolated from natural sources are presented as potent scavengers of ROS. ROS scavenging is an alternative way to protect destruction of cell components initiated by free radicals. The paper is composed very clearly, but I have found few errors which should be remove.
In figure 2, lipid peroxidation; in the second and fourth equation the left side in unequal to right one.
In figure 4, the chemical structures of piperazine-erastin and imidazole ketone erastin must be change in place.
In page 8, line 242 round bracket should be erased and added in line 240.
In page 8 line 247 the sentence” concentration of” is not terminate.
In figure 6 iron chelator D has proper name Liproxstatin-1
In Table 1, section antioxidant activities, line 4 should be Fenton reaction (it is specific name).
In reference 2, it should be fully developed to J R Soc Med. 2003 Jan; 96(1): 46–47. I must mention that this reference is based on a book of Lane N. “Oxygen—the Molecule that made the World”. Oxford: Oxford University Press, 2002. [384 pp; ISBN 0-19-850803-4.
The following paper can be add as a reference: Ferroptosis-Specific Inhibitor Ferrostatin-1 Relieves H2O2-Induced Redox Imbalance in Primary Cardiomyocytes through the Nrf2/ARE Pathway; Chaofeng Sun, Fang Peng, Jianfei Li, Xudong Cui, Xin Qiao, Wangliang Zhu; https://doi.org/10.1155/2022/4539932
Author Response
Thank you for your words. Your reviewed points are taken into consideration and answered as follows in bold.
In figure 2, lipid peroxidation; in the second and fourth equation the left side in unequal to right one.
The equations are corrected.
In figure 4, the chemical structures of piperazine-erastin and imidazole ketone erastin must be change in place.
Done.
In page 8, line 242 round bracket should be erased and added in line 240.
Done.
In page 8 line 247 the sentence” concentration of” is not terminate.
The sentence is corrected: “concentration of cystine”.
In figure 6 iron chelator D has proper name Liproxstatin-1
The name is corrected.
In Table 1, section antioxidant activities, line 4 should be Fenton reaction (it is specific name).
Done.
In reference 2, it should be fully developed to J R Soc Med. 2003 Jan; 96(1): 46–47. I must mention that this reference is based on a book of Lane N. “Oxygen—the Molecule that made the World”. Oxford: Oxford University Press, 2002. [384 pp; ISBN 0-19-850803-4.
This reference has been corrected in the manuscript as following: 2. Carter, R. Oxygen: The Molecule That Made the World. J. R. Soc. Med. 2003, 96(1), 46–47. Based on: Lane N. Oxygen—the Molecule that made the World. Oxford: Oxford University Press, 2002. [384 pp; ISBN 0-19-850803-4]
The following paper can be add as a reference: Ferroptosis-Specific Inhibitor Ferrostatin-1 Relieves H2O2-Induced Redox Imbalance in Primary Cardiomyocytes through the Nrf2/ARE Pathway; Chaofeng Sun, Fang Peng, Jianfei Li, Xudong Cui, Xin Qiao, Wangliang Zhu; https://doi.org/10.1155/2022/4539932
This reference has been added on page 13 from line 386 to line 387 within this sentence “Fer-1 was shown to inhibit cell death and to restore redox balance in a cell model of ischemic heart disease induced by H2O2”
Reviewer 2 Report
Dear authors,
Ferroptosis is a promising target for prevention and treatment of wide range of diseases as atherosclerosis, Alzheimer, diabetes, cancer, and renal failure. I appreciate the hard work and the effort spent in writing this review. Shedding the lights on this type of cell death can inspire other researchers for further study on its mechanisms and trying to find other active entities that can be used in treatment and prevention of variouse diseases. However, before getting published in Molecules, the following comments should be taken in consideration.
Line 93: “Notably, iron excess participates to lipid peroxidation” participate to is not grammatically correct and should be participate in.
Line 95: “Fe2+ is oxidized in Fe3+ ” should be into or to
Line 204: “Acyl-Co1 synthase long-chain family 4 (ACSL4)” Do you mean Acyl-CoA Synthetase Long Chain Family Member 4? Please correct in the manuscript and the figure.
Author Response
Thank you for your words. Your comments are taken into consideration and answered as follows in bold.
Line 93: “Notably, iron excess participates to lipid peroxidation” participate to is not grammatically correct and should be participate in.
The sentences is corrected.
Line 95: “Fe2+ is oxidized in Fe3+ ” should be into or to
Done.
Line 204: “Acyl-Co1 synthase long-chain family 4 (ACSL4)” Do you mean Acyl-CoA Synthetase Long Chain Family Member 4? Please correct in the manuscript and the figure.
The name is corrected.
Reviewer 3 Report
The examination of the manuscript ID: Molecules-2213436, submitted for our consideration, raises an interesting question and this study is worthy of interest. The title is very clear in our opinion: Nature-inspired Bioactive Compounds: A Promising Approach for Ferroptosis-linked Human diseases? The title is very clear in our opinion, however, the review discusses the antioxidant properties of natural compounds that have promising effects on ferroptosis-related conditions, but not, in our opinion, the action of bioactive peptides on ferroptosis regulation. This should also appear in this study. In doing so, the conclusion remains incomplete if we stick to the title of the manuscript.
1- The authors should improve the section: action of bioactive peptides on the regulation of ferroptosis or delete this part if there are no clear data.
2- The authors should therefore revise the conclusion to make it more complete and comprehensive.
Author Response
Thank you for your words. Your comments are taken into consideration and answered as follows in bold.
- The authors should improve the section: action of bioactive peptides on the regulation of ferroptosis or delete this part if there are no clear data.
As we mentioned page 15 from line 488 to line 491 “Although these peptides are widely reported for their antioxidant activities such as radical scavenging, reducing power, and metal chelation, to date, there is no study investigating them in a ferroptosis model”, up to now, there is no clear evidence of anti-ferroptosis activity of those bioactive peptide.
However, in this chapter, we wanted to highlight the potential and promising approach to use those peptides showing antioxidant, metal chelation capacity and also influence on ferroptosis regulation biomarkers (GSH, ROS production, lipid peroxidation, and antioxidant enzymes (SOD, CAT and GPx) activities) in a ferroptotic environment. We here wanted to provide constructive recommendations for future research dealing with ferroptosis as a review article should do. In that spirit, to advise properly the lecturer we added on p15 from line 491 to line 493, this sentence to warn the lecturer “Therefore, in this part we will review the antioxidant properties of bioactive peptides such as metal chelation, reduction of lipid peroxidation, modulation of GSH content and GPx activity that might confer them interesting for anti-ferroptotic properties.”
2- The authors should therefore revise the conclusion to make it more complete and comprehensive.
The conclusion has been revised as follows:
In summary, the discovery of ferroptosis was a huge scientific progress in the biological domain. First mainly used as a tool to kill cancer cells, ferroptosis was then discovered as implicated in cardiovascular and neurodegenerative diseases. The great value of identifying ferroptosis inducers and inhibitors was translated in diseases-treatment studies. However, up to date, ferroptosis have not been reported in any clinical trial (www.clinicaltrials.gov). Yet, more in-depth studies are required to make connections between cell/animal-based data and the complexity of the human organism. The main problem might be the lack of cell-based ferroptosis model related to those diseases to evaluate ferroptosis inhibitors. Nevertheless, scientists have also addressed people’s need to consume natural products by studying multiple different natural compounds on ferroptosis regulation. Although bioactive peptides have not been directly linked to ferroptosis regulation, their acquire various antioxidant activities such as iron chelation properties, modulation of ferroptosis biomarkers and restoration of redox homeostasis might open a new area for the evaluation of those natural sources on ferroptosis-related conditions.